# Revisiting the Initial Steps in Adaptive Gradient Descent Optimization

Abulikemu Abuduweili , Changliu Liu
Robotics Institute, Carnegie Mellon University
abulikea@andrew.cmu.edu, cliu6@andrew.cmu.edu

Adaptive gradient optimization methods, such as Adam, are prevalent in training deep neural networks across diverse machine learning tasks due to their ability to achieve faster convergence. However, these methods often suffer from suboptimal generalization compared to stochastic gradient descent (SGD) and exhibit instability, particularly when training Transformer models. In this work, we show the standard initialization of the second-order moment estimation ($v_0 = 0$) as a significant factor contributing to these limitations. We introduce simple yet effective solutions: initializing the second-order moment estimation with non-zero values, using either data-driven or random initialization strategies. Empirical evaluations demonstrate that our approach not only stabilizes convergence but also enhances the final performance of adaptive gradient optimizers. Furthermore, by adopting the proposed initialization strategies, Adam achieves performance comparable to many recently proposed variants of adaptive gradient optimization methods. Our code is available at `https://github.com/Walleclipse/Adam_Initialization`.

## 1. Introduction

First-order optimization methods, such as stochastic gradient descent (SGD), have been foundational in training deep neural networks due to their robust convergence properties across various applications [1]. However, as deep learning architectures have grown more complex, there has been increasing interest in adaptive gradient optimizers, which dynamically adjust learning rates based on the gradients of individual parameters [2]. These methods often lead to faster convergence in certain tasks [3]. Among them, Adam has emerged as one of the most widely used adaptive gradient methods, successfully applied to fields such as computer vision, natural language processing, and reinforcement learning [4]. By combining the benefits of momentum and adaptive learning rates, Adam has proven particularly effective in training generative models and large language models [5]. Its success is particularly evident in transformer-based architectures, where careful hyperparameter tuning or the use of a learning rate warmup strategy has enabled state-of-the-art performance [6–8]. Additionally, theoretical studies have provided insights into Adam's convergence properties in non-convex settings, further solidifying its utility [9].

At the same time, Adam's effectiveness is not without limitations. While it is known for its fast convergence and adaptability, it can exhibit instability and poor generalization in specific non-convex optimization. For example, in training transformers for language models, the omission of learning rate warmup strategies has been linked to instability and suboptimal generalization [10, 11]. These instabilities often lead the optimizer to converge to suboptimal local minima, undermining model performance. To address these challenges, several modifications to Adam have been proposed. For instance, AdaBound [12] improves generalization by bounding the step size with a smooth parameter update, while RAdam [11] rectifies the variance of the second-order moment to stabilize the learning rate during early iterations. AdaBelief [13] adapts the step size based on the "belief" in the observed gradients, enhancing generalization. A broader range of studies has introduced further refinements to stabilize convergence and improve generalization performance [14–16]. Additionally, the warmup heuristic, which employs a small learning rate during the initial training epochs, has been adopted to improve stability and generalization in Adam [17].

Second Conference on Parsimony and Learning (CPAL 2025).

The update rule of Adam can be understood as a combination of update direction, determined by the sign of the stochastic gradients, and update magnitude [18]. Recent works have explored the role of Sign Gradient Descent (SignGD) as a surrogate for understanding Adam's behavior [19, 20]. We identify a critical factor contributing to Adam's instability: its default initialization of the second-order moment estimation ($v_0 = 0$), which causes Adam to exhibit sign-descent behavior in its initial steps. This default setting introduces high variance in the second-moment estimation and update step size, resulting in unstable convergence, particularly during the early stages of training. This instability often prevents the optimizer from reaching well-generalized optima. To address this issue, we propose a simple yet effective modification: initializing the second-order moment estimation with non-zero values. These initial values can be derived from data-driven statistics of squared gradient, or even assigned as random positive numbers. This modification reduces the variance of the second moment and stabilizes the optimization process. Our empirical evaluations across a wide range of tasks demonstrate that the proposed initialization of the second-order moment significantly improves the stability and overall performance of adaptive gradient optimizers, particularly in non-convex settings. The contributions of this paper are as follows:

- We show that the zero initialization of the second-order moment is a significant factor contributing to Adam's instability.

- We propose a simple yet effective modification: initializing the second-order moment estimation with data-driven or random non-zero values to enhance the stability and performance of adaptive gradient methods.

- Through extensive experiments, we demonstrate that the proposed initialization strategy of $v_0$ improves the performance of several adaptive gradient methods.

## 2. Second-order Moment Initialization of Adam

This section focuses on the instability in the Adam optimizer caused by the standard zero-initialization of the second-order moment. Unlike the non-convergence issues discussed in prior works [10], the instability we address primarily affects the early stages of optimization in non-convex problems, particularly in deep neural networks. While this issue has minimal impact on convex problems, it can significantly hinder optimization in more complex, non-convex landscapes.

### 2.1. Revisiting the Adam Optimizer

**Update rule of Adam.** The update rule for Adam is given by the following equations [4]:

$$m_t = \beta_1 m_{t-1} + (1 - \beta_1) g_t = \beta_1^t m_0 + (1 - \beta_1) \sum_{k=0}^{t-1} \beta_1^k g_{t-k}, \ \hat{m}_t = \frac{m_t}{1 - \beta_1^t} \tag{1}$$

$$v_t = \beta_2 v_{t-1} + (1 - \beta_2) g_t^2 = \beta_2^t v_0 + (1 - \beta_2) \sum_{k=0}^{t-1} \beta_2^k g_{t-k}^2, \ \hat{v}_t = \frac{v_t}{1 - \beta_2^t} \tag{2}$$

$$\theta_t = \theta_{t-1} - \alpha \frac{\hat{m}_t}{\sqrt{\hat{v}_t} + \epsilon} \tag{3}$$

where $m_t$ and $v_t$ represent the first and second moments, $g_t$ is a gradient of objective function. $\beta_1, \beta_2$ are the decay rates for the first and second-moment estimates, $\alpha$ is the learning rate, and $\epsilon$ is a small constant preventing division by zero. We rewrite the above term to illustrate the sign, and magnitude of the Adam [18]. Ignoring $\epsilon$, since it is very small in practice, we have the step size:

$$\Delta \theta_t = \theta_t - \theta_{t-1} = -\alpha \frac{\hat{m}_t}{\sqrt{\hat{v}_t}} = -\alpha \sqrt{\frac{1}{1 + \frac{\hat{v}_t - \hat{m}_t^2}{\hat{m}_t^2}}} \cdot \text{sign}(\hat{m}_t) \tag{4}$$

**First step of Adam as sign descent.** In Adam's standard implementation, the first- and second-order momentum terms are initialized to zero, $m_0 = 0, v_0 = 0$. As a result, the first step of the

optimization process degenerates into sign descent, where the magnitude of the step size depends solely on the learning rate $\alpha$ rather than the full gradient. This behavior is illustrated as follows:

$$\Delta\theta_1 = -\alpha \frac{g_1}{\sqrt{g_1^2 + \frac{\beta_2}{1-\beta_2} v_0}} = -\alpha \cdot \text{sign}(g_1). \tag{5}$$

In this first step, Adam performs a pure sign-descent update due to the zero initialization of $m_0 = 0, v_0 = 0$. However, from the second step onward, the moving averages begin to incorporate gradient information, and the updates evolve into a combination of sign descent and adaptive gradient descent. Over subsequent iterations, as more gradient information is accumulated, the influence of the initial sign descent diminishes, and the optimizer transitions into its adaptive behavior where $m_t \neq v_t$ , as shown in Equations (1), (2) and (4).

## 2.2. Instability of Adam optimizer

**Instability of Adam on training Transformer network.** Training Transformer models for various NLP tasks often relies on a learning rate warmup strategy [21], which has also been shown to enhance accuracy in Vision Transformers [22, 23]. Removing the warmup phase, however, has been observed to increase training loss, underscoring its role in stabilizing the optimization process [11].

To explore this phenomenon, we conducted experiments training a Transformer model on the IWSLT'14 DE-EN dataset for a neural machine translation task. We evaluated three approaches: vanilla Adam without warmup (denoted as $v_{0,0}$), vanilla Adam with warmup, and our proposed data-driven initialization of Adam without warmup (denoted as $v_{0,data}$, described in the next section). As illustrated in Figure 1(a), vanilla Adam without warmup exhibits increased training loss during the early stages. We attribute this instability to Adam's initial sign-descent behavior, which is exacerbated by the standard zero-initialization of the second-order moment ($v_0 = 0$). While the learning rate warmup strategy effectively addresses this issue, it requires using a very small learning rate during the initial stages, limiting parameter updates and slowing down convergence. In this work, we propose a non-zero initialization strategy to directly stabilize the optimizer. Unlike warmup, our approach avoids restrictive learning rate constraints, enabling faster convergence while maintaining training stability.

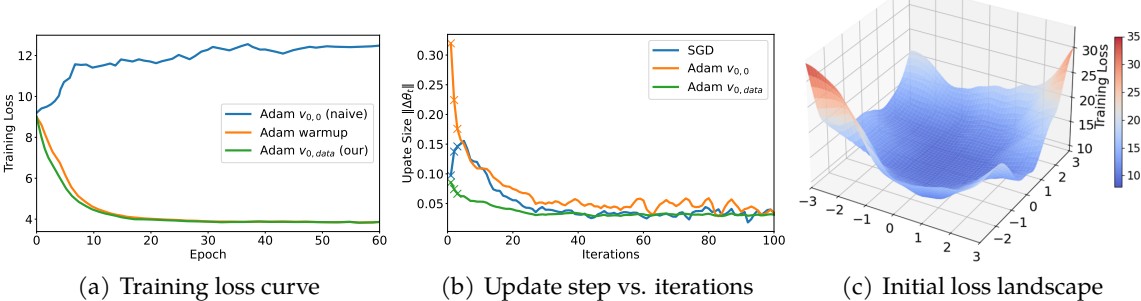

(a) Training loss curve     (b) Update step vs. iterations     (c) Initial loss landscape

Figure 1: Training Transformers on the IWSLT'14 De-En dataset.

**Impact of sign descent and shrinking gradients.** In this section, we analyze the non-convergence behavior of vanilla Adam, focusing on the large initial step sizes observed during neural network training (Figure 1(b)). Neural networks often exhibit a flat loss landscape at the beginning of training, with gradients that are small in magnitude. This phenomenon is particularly pronounced when training Transformers, as noted in prior works [24–26]. The initial loss landscape of the Transformer model is visualized in Figure 1(c), where the loss is plotted along two random directions as described in [27]. The visualization highlights that the loss landscape is extremely flat, and gradients are correspondingly small. When training such networks with Adam, the "sign descent" behavior during the initial step can amplify these small gradients disproportionately, resulting in overly large parameter updates. To further investigate this phenomenon, Figure 1(b) illustrates the norm of the update step $\|\Delta\theta_t\|$ during training for three optimizers: SGD, vanilla Adam, and Adam with the

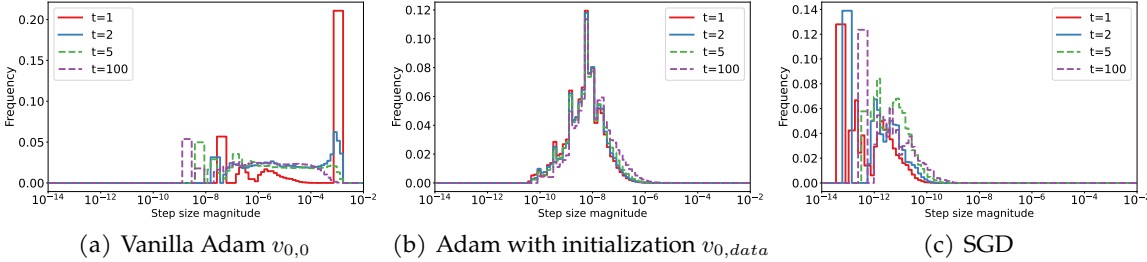

(a) Vanilla Adam $v_{0,0}$      (b) Adam with initialization $v_{0,data}$      (c) SGD

Figure 2: Histogram of update step distribution across coordinates.

proposed initialization $v_{0,data}$. The results show that the first update step size for vanilla Adam $v_{0,0}$ is significantly larger compared to Adam $v_{0,data}$ or SGD. These large initial updates can push the optimizer away from initial regions in the parameter space, making recovery and convergence more challenging. In contrast, SGD exhibits much smaller update steps during the initial stages, even when using a larger learning rate (lr=0.1) than Adam (lr=0.001) in our experiments. To further illustrate the update step sizes, Figure 2 presents histograms of the absolute values of parameter updates for different optimizers. For vanilla Adam (Figure 2(a)), many parameters are updated with a step size equal to the learning rate in the first step ($t = 1$) due to its "sign descent" behavior. Subsequently, the update step sizes decrease. In contrast, Adam with non-zero initialization (Figure 2(b)) achieves relatively stable update step sizes throughout training, avoiding the large initial jumps seen in vanilla Adam. This behavior aligns closely with SGD (Figure 2(c)), which consistently maintains stability in its updates from the start.

## 2.3. Non-zero Initialization of Second Order Moment

As shown in Equations (2) and (4), initializing the second-order moment $v_0$ with non-zero values effectively prevents the first step of Adam from degenerating into sign descent.

**Special case: linear loss.** To build intuition for initializing the second-order moment, we first study a simplified setting. Consider the linear loss function $f(\theta_t) = \langle \theta_t, g_t \rangle$ with a Noisy Gradient Oracle with Scale Parameter (NGOS), a widely used framework for analyzing training dynamics of optimizers [28, 29]. In this setting, the stochastic gradient $g_t$ is sampled from a Gaussian distribution with mean $\bar{g}$ and and variance $\sigma^2 I$, i.e. $g_t \sim \mathcal{N}(\bar{g}, \sigma^2 I)$. This setup mimics mini-batch training in neural networks, where the stochastic gradient is provided as a noisy approximation of the full gradient. Using this framework, the expectation of first- and second-order moments is given by

$$\mathbf{E}[m_t] = \beta_1^t m_0 + (1 - \beta_1) \sum_{k=0}^{t-1} \beta_1^k \bar{g} = \beta_1^t m_0 + (1 - \beta_1^t) \bar{g} \tag{6}$$

$$\mathbf{E}[v_t] = \beta_2^t v_0 + (1 - \beta_2) \sum_{k=0}^{t-1} \beta_2^k (\bar{g}^2 + \sigma^2 I) = \beta_2^t v_0 + (1 - \beta_2^t)(\bar{g}^2 + \sigma^2 I) \tag{7}$$

These results indicate that, after a sufficient number of steps, $\mathbf{E}[m_t] \approx \bar{g}$, and $\mathbf{E}[v_t] \approx \bar{g}^2 + \sigma^2 I$. In many practical scenarios, where the average gradient magnitude is small $\mathbf{E}[g_t] \approx 0$, initializing $m_0 = 0$ is a reasonable choice to stabilize $m_t$. Since $m_t$ approximates the first moment of the gradient, zero initialization aligns with its role. However, for $v_t$, which represents the second-order moment of the gradient, it must satisfy $\mathbf{E}[v_t] > 0$. This makes the standard zero initialization ($v_0 = 0$) inherently inconsistent with its purpose. Furthermore, $v_0$ plays a critical role in determining the adaptive learning rate during the initial steps, directly influencing convergence and optimization stability.

To assess the stability of the optimization process and the influence of the initial state, we define the drift of the second-order moment as:

$$\text{drift}_{v_t}(v_0) = \|\mathbf{E}[v_\infty] - \mathbf{E}[v_0]\|. \tag{8}$$

This term quantifies the adjustment required for the second moment to transition from its initial value to its steady-state. It reflects how much the optimizer must adapt its gradient scaling during training. Since $v_t$ directly determines the adaptive learning rate, a smaller drift term indicates better stability of optimization process.

For vanilla Adam, $v_0 = 0$, the expected value of $v_t$ converges to $\mathbf{E}[v_\infty]| = \bar{g}^2 + \sigma^2 I$ from $\mathbf{E}[v_0] = 0$. Then $\text{drift}_{v_t}(v_0 = 0) = \bar{g}^2 + \sigma^2$. This large drift value causes significant initial adjustments of $v_t$, leading to potential instability in optimization.

For non-zero initialization, $v_0 = \bar{g}^2 + \sigma^2 I$, the expected second moment remains constant for all $\mathbf{E}[v_t] = \bar{g}^2 + \sigma^2 I$. Thus $\text{drift}_{v_t}(v_0 = \bar{g}^2 + \sigma^2 I) = 0$. With this initialization, $v_t$ is immediately aligned with its steady-state value, eliminating the need for adjustments and ensuring stability from the start. The expectation $\mathbf{E}[v_t]$ is of scale $\mathcal{O}(\sigma^2)$ and the standard deviation of each coordinate of $v_t$ is of scale $\mathcal{O}((1-\beta_2)\sigma^2)$. When $\beta_2$ is close to 1, $v_t$ becomes nearly deterministic and tightly concentrates around $v_t \approx \bar{g}^2 + \sigma^2 I$. Ignoring $\epsilon$ for simplicity, the Adam update rule becomes:

$$\theta_t \approx \theta_{t-1} - \alpha \frac{m_t}{\sqrt{\bar{g}^2 + \sigma^2 I}} \tag{9}$$

This ensures a stable adaptive learning rate: $\alpha \cdot (\bar{g}^2 + \sigma^2 I)^{-1/2}$. Such stability aligns with the definition of an adaptive learning rate, where $v_t$ incorporates local geometry (e.g., Hessian information). For the linear loss case, this stability results in more consistent updates. Further illustration of the stability provided by a non-zero $v_0$ in RMSprop is presented in Appendix A.1.

For random initialization, $v_0 = \lambda I, \lambda > 0$, the the drift term becomes: $\text{drift}_{v_t}(v_0 = \lambda I) = |\bar{g}^2 + \sigma^2 - \lambda|$. For any $0 < \lambda < 2(\bar{g}^2 + \sigma^2)$, this drift term is smaller than that of zero initialization: $\text{drift}_{v_t}(v_0 = \lambda I) < \text{drift}_{v_t}(v_0 = 0)$. This reduced drift results in a more stable optimization process compared to $v_0 = 0$, even with random initialization.

**Initialization of $v_0$.** Inspired by the analysis of linear loss cases with stochastic gradients, we propose two different non-zero initialization strategies for the second-order moment $v_0$.

- *Data-driven Initialization*, denoted as $v_{0,data}$. In the data-driven strategy, $v_0$ is initialized using the gradient statistics calculated from sampled training data $(x_i, y_i) \sim \mathcal{D}$, where $\mathcal{D}$ represents the training set. Specifically, for sampled data $(x_i, y_i)$, the gradient of the loss function is computed as: $g(x_i, y_i) = \nabla_\theta f(x_i, y_i)$ for $(x_i, y_i)$. The second-order moment is then initialized as:

$$v_0 = \sigma \cdot \left( \mathbf{E}[g(x_i, y_i)]^2 + \mathbf{VAR}[g(x_i, y_i)] \right), \text{ where } (x_i, y_i) \sim \mathcal{D}. \tag{10}$$

  Here, $\sigma$ is a hyperparameter that controls the scale of $v_0$. This approach ensures that $v_0$ reflects meaningful statistical information about the gradient, aligning the optimizer's initialization with the characteristics of the specific training data.

- *Random Initialization*, denoted as $v_{0,rnd}$. This is computationally efficient and avoids the overhead associated with data-driven initialization. As shown in the previous analysis, any small positive value for $v_0$ enhances the stability of $v_t$, making random initialization a practical choice. We propose initializing $v_0$ using a scaled chi-squared distribution [1]:

$$v_0 \sim \frac{\sigma}{\text{fan}_{\text{in}} + \text{fan}_{\text{out}}} \cdot \chi_1^2, \tag{11}$$

  where $\chi_1^2$ denotes a chi-squared distribution with one degree of freedom. $\text{fan}_{\text{in}}$ and $\text{fan}_{\text{out}}$ are the input and output dimensions of the weight matrix $\theta \in \mathcal{R}^{\text{fan}_{\text{out}} \times \text{fan}_{\text{in}}}$, and $\sigma$ is a hyperparameter that controls the scale of the distribution. This distribution ensures that $v_0$ scales appropriately with the dimensions of the weight parameters, similar to Xavier initialization for neural network weights [30]. Furthermore, the squared value $g_t^2$ of a Gaussian random gradient $g_t$ naturally follows a scaled chi-squared distribution, providing a principled foundation for this initialization strategy. Please refer to Appendix B for the pseudocode of the proposed initialization methods. Note that only weight matrices with two or more dimensions are initialized.

---

[1]Which is also can be described as gamma distribution, $v_0 \sim \text{Gamma}\left(\frac{1}{2}, \frac{2(\text{fan}_{\text{in}} + \text{fan}_{\text{out}})}{\sigma}\right)$

Under the proposed initialization $v_{0,data}$ and $v_{0,rnd}$, the first step size of Adam becomes:

$$\Delta\theta_1 = -\alpha \frac{g_1}{\sqrt{g_1^2 + \frac{\beta_2}{1-\beta_2}v_0}} \neq -\alpha \cdot \text{sign}(g_1), \; |\Delta\theta_1| < \alpha \tag{12}$$

This ensures that the first update step is influenced by both the magnitude and direction of the gradient, avoiding the pure "sign descent" behavior seen with $v_0 = 0$. Such stabilization is particularly crucial for deep learning tasks with shrinking gradients, such as training Transformers. The proposed initialization strategies are broadly applicable beyond Adam and can be extended to other adaptive gradient methods, including AMSGrad [10, 31], AdaBound [12], RAdam [11], and AdaBelief [13]. These methods could benefit from improved stability during the initial steps, potentially enhancing both training dynamics and final performance. A discussion comparing the proposed initialization strategy with other optimization approaches is presented in Appendix A.2.

## 3. Experiments

To evaluate the effectiveness of our approach, we conducted extensive experiments across a variety of tasks, including image classification with convolutional neural networks (CNNs) [32], image generation with generative adversarial networks (GANs) [33], language modeling with long short-term memory networks (LSTMs) [34], and neural machine translation with Transformers [17]. We empirically evaluate the performance of two initialization strategies — $v_{0,data}$ (Equation (10)) and $v_{0,rnd}$ (Equation (11)) — across several widely used adaptive gradient optimization methods. These methods include SGD with momentum [35, 36], Adam [4], AdamW [37], AdaBound [12], RAdam [11], and AdaBelief [13]. For each optimizer, we use the standard initialization ($v_0 = 0$) as the baseline and compare it against the proposed strategies ($v_{0,rnd}$ and $v_{0,data}$). For $v_{0,data}$, gradient statistics are computed using 5,000 random samples prior to training, with the scaling factor set to $\sigma = 1$. For $v_{0,rnd}$, the scaling factor is set to $\sigma = 100$. Detailed information about the experimental setup is provided in Appendix C.1.

### 3.1. Toy Experiments of Adam's Instability and Initialization

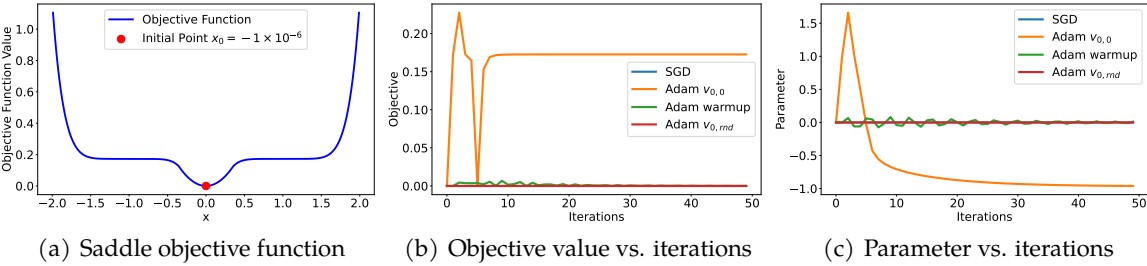

(a) Saddle objective function     (b) Objective value vs. iterations     (c) Parameter vs. iterations

Figure 3: Optimization of the saddle objective function with different methods.

We conduct a toy experiment to illustrate the instability of Adam with its standard zero initialization and the effectiveness of our proposed non-zero initialization. For this demonstration, we use the random initialization strategy $v_{0,rnd}$. The objective function is a non-convex saddle function:

$$f(x) = \begin{cases} (x-b)^n, & \text{if } x \geq x_s \\ -(x+b)^n, & \text{if } x \leq -x_s \\ x^2 + d, & \text{if } -x_s < x < x_s \end{cases} \tag{13}$$

Here $x_s$ is a switch point, $b$ is a bias and $d$ is a shift ensuring smooth transition at the switch points.

$$x_s = \left(\frac{s}{n}\right)^{\frac{1}{n-1}} + b, \; d = (x_s - b)^n - x_s^2 \tag{14}$$

The parameter $n$ represents the degree of the polynomial. In our experiment, we set $n = 7$, $b = 1$, and $s = 0.5$. The purpose of the experiment is to observe the optimization behavior

under different initializations. We use the Adam optimizer with the following hyperparameters: $\alpha = 1, \beta_1 = 0.9, \beta_2 = 0.999$. For scenarios requiring smaller learning rates, the objective function can be scaled down to achieve similar conclusions.

The optimization process starts at an initial point $x_0 = -10^{-6}$, close to the true optimum $x^\star = 0$, as shown in Figure 3(a). Figure 3(b) and Figure 3(c) depict the loss values and parameter convergence over iterations for different methods. Standard Adam with $v_0 = 0$ converges to a suboptimal local minimum around $x_\infty \approx -1$, far from the true optimum. In contrast, Adam with the proposed non-zero initialization $v_{0,\text{rnd}}$ converges successfully to the true optimum. As a baseline, both the SGD and Adam with warmup also converge near the optimum; however, the proposed method $v_{0,\text{rnd}}$ demonstrates greater stability and efficiency, as reflected in the convergence values in Appendix C.2.

## 3.2. Image Classification with CNN

We evaluate the ResNet-34 architecture [32] on the CIFAR-10 image classification dataset [38]. The test accuracy at the final epoch is summarized in Table 1. The results demonstrate that the proposed initialization of $v_0$, represented as $v_{0,rnd}$ and $v_{0,data}$, enhances the performance of adaptive gradient optimization methods, including Adam, AdamW, AdaBound, RAdam, and AdaBelief. Notably, with $v_{0,data}$, Adam achieves a test accuracy surpassing that of the more recent AdaBelief approach. Furthermore, AdaBelief with $v_{0,data}$ outperforms SGD, showcasing the effectiveness of the proposed method. $v_{0,rnd}$ also consistently improves the performance of adaptive gradient methods without incurring additional computational overhead, making it a practical and efficient solution for stabilizing the optimization process.

Table 1: Test accuracy ↑ (%) of ResNet-34 on CIFAR-10 dataset.

| Optimization | SGD | Adam | AdamW | AdaBound | RAdam | AdaBelief |
|---|---|---|---|---|---|---|
| Vanilla $v_{0,0}$ | 96.19±0.09 | 95.25±0.11 | 95.36±0.11 | 95.38±0.07 | 95.61±0.16 | 95.94±0.07 |
| $v_{0,rnd}$ | - | 95.87±0.09 | 95.94±0.09 | 95.80±0.07 | 95.83±0.11 | 96.11±0.07 |
| $v_{0,data}$ | - | 96.02±0.09 | 95.95±0.09 | 95.96±0.07 | 95.90±0.12 | **96.24±0.07** |

To further validate the effectiveness of our algorithm on a more comprehensive dataset, we conducted experiments on the ImageNet dataset [39], utilizing ResNet-18 as the backbone network. As shown in Table 2, both $v_{0,rnd}$ and $v_{0,data}$ provide significant performance gains across several adaptive gradient optimization methods.

Table 2: Test accuracy ↑ (%) of ResNet-18 on ImageNet dataset.

| Optimization | SGD | Adam | AdamW | AdaBound | RAdam | AdaBelief |
|---|---|---|---|---|---|---|
| Vanilla $v_{0,0}$ | 70.23±0.07 | 63.79±0.12 | 67.93±0.12 | 68.13±0.11 | 67.62±0.11 | 70.08±0.10 |
| $v_{0,rnd}$ | - | 65.99±0.11 | 68.95±0.11 | 68.80±0.11 | 68.83±0.11 | 70.69±0.10 |
| $v_{0,data}$ | - | 66.13±0.11 | 68.49±0.11 | 68.96±0.11 | 68.99±0.11 | **70.77±0.10** |

## 3.3. Language Modeling with LSTM

We evaluate a 2-layer LSTM network [34] on the language modeling task of Penn Treebank dataset [40]. The test perplexity (lower is better) is summarized in Table 3. The results demonstrate that both $v_{0,rnd}$ and $v_{0,data}$ significantly improve the performance of adaptive gradient methods. Notably, with these proposed initialization strategies, Adam achieves test perplexity results that surpass the more recent AdaBelief optimizer. Results for a 3-layer LSTM network are provided in Appendix C.3.

Table 3: Test perplexity ↓ of 2 Layer LSTM on Penn Treebank dataset.

| Optimization | SGD | Adam | AdamW | AdaBound | RAdam | AdaBelief |
|---|---|---|---|---|---|---|
| Vanilla $v_{0,0}$ | 67.25±0.20 | 67.11±0.20 | 73.61±0.15 | 67.69±0.24 | 73.61±0.25 | 66.75±0.11 |
| $v_{0,rnd}$ | - | 66.70±0.17 | 68.35±0.14 | 66.94±0.19 | 68.55±0.17 | 66.12±0.10 |
| $v_{0,data}$ | - | 66.37±0.17 | 69.31±0.14 | 66.90±0.19 | 69.32±0.17 | **65.87±0.10** |

### 3.4. Neural Machine Translation with Transformer

We evaluated a small Transformer model [17] using the Fairseq package [41] on the IWSLT'14 German-to-English machine translation dataset. The BLEU scores [42] are summarized in Table 4. The results demonstrate that the proposed initialization strategies, $v_{0,rnd}$ and $v_{0,data}$, provide significant performance improvements for adaptive gradient optimization methods.

Table 4: BLEU score $\uparrow$ of Transformer on IWSTL'14 DE-EN dataset.

| Optimization | SGD | Adam | AdamW | RAdam | AdaBelief |
|---|---|---|---|---|---|
| Vanilla $v_{0,0}$ | 28.22±0.21 | 30.14±0.39 | 35.62±0.11 | 34.76±0.14 | 35.60±0.11 |
| $v_{0,rnd}$ | - | 33.71±0.19 | 36.06±0.11 | 34.97±0.14 | 36.12±0.11 |
| $v_{0,data}$ | - | 33.64±0.20 | 35.98±0.11 | 34.84±0.14 | **36.18±0.11** |

### 3.5. Image Generation with GAN

We evaluated a deep convolutional GAN (DCGAN) [43] on the CIFAR-10 image generation task. The performance is measured using the Frechet Inception Distance (FID, lower is better) [44], which quantifies the similarity between generated images and the real dataset. In training GANs, optimizer stability is crucial for achieving high-quality image generation. As shown in Table 5, the proposed initialization strategies, $v_{0,rnd}$ and $v_{0,data}$, stabilize the optimization process for adaptive gradient methods, resulting in additional performance gains. For instance, $v_{0,rnd}$ and $v_{0,data}$ improve the performance of the Adam optimizer by 10% and 13%, respectively, highlighting the effectiveness of the proposed approaches.

Table 5: FID score $\downarrow$ of GAN on CIFAR-10 dataset dataset.

| Optimization | SGD | Adam | AdamW | AdaBound | RAdam | AdaBelief |
|---|---|---|---|---|---|---|
| Vanilla $v_{0,0}$ | 237.77±147.9 | 54.22±4.21 | 52.39±3.62 | 118.75±40.64 | 48.24±1.38 | 47.25±0.79 |
| $v_{0,rnd}$ | - | 48.60±3.19 | 46.94±3.21 | 92.36±35.76 | 47.70±1.32 | 45.91±0.78 |
| $v_{0,data}$ | - | 47.02±3.20 | 45.25±3.07 | 85.45±36.31 | 47.84±1.24 | **45.02±0.78** |

### 3.6. Further Discussion of the Proposed Initialization Method

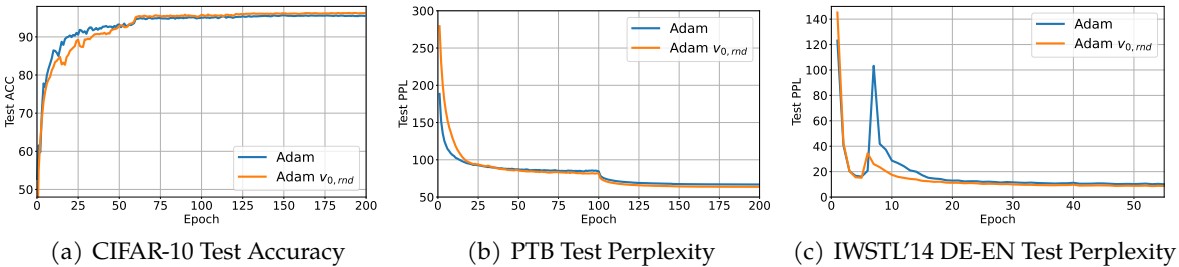

(a) CIFAR-10 Test Accuracy     (b) PTB Test Perplexity     (c) IWSTL'14 DE-EN Test Perplexity

Figure 4: Comparison of Vanilla Adam and Adam $v_{0,rnd}$ on (a) CIFAR-10 image classification task. (b) Penn Treebank language modeling task. (c) IWSTL'14 machine translation task.

**Training curve.** We compare the training curves of Vanilla Adam and Adam with random initialization $v_{0,rnd}$, as it is more computationally efficient. In the CIFAR-10 image classification task in Figure 4(a), while Adam $v_{0,rnd}$ exhibits slightly lower accuracy in the initial steps, it achieves more stable convergence and higher final accuracy. For the Penn Treebank language modeling task in Figure 4(b), Adam $v_{0,rnd}$ results in lower perplexity at convergence compared to Vanilla Adam. For Transformer models on the IWSLT'14 DE-EN machine translation dataset (with warmup) in Figure 4(c), Adam $v_{0,rnd}$ demonstrates faster convergence, more stable optimization, and lower perplexity at the end of training.

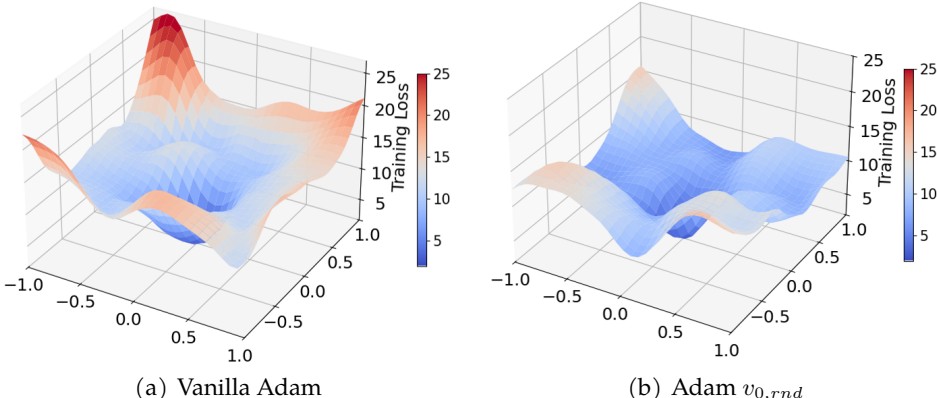

(a) Vanilla Adam                              (b) Adam $v_{0,rnd}$

Figure 5: Comparison of the loss landscape around the convergent points of Transformer trained by vanilla Adam and Adam $v_{0,rnd}$.

**Loss landscape.** To further explore the converged behavior of Adam with $v_{0,rnd}$, we visualize the loss landscapes around the convergent points of Transformer models trained with Vanilla Adam and Adam $v_{0,rnd}$ on the IWSLT'14 DE-EN machine translation task. The loss landscape is plotted along two normalized random directions. As shown in Figure 5, the loss landscape for Adam $v_{0,rnd}$ is flatter than that for Vanilla Adam. A flatter loss landscape is often indicative of better generalization performance [45, 46]. Although the training losses of Vanilla Adam and Adam $v_{0,rnd}$ are comparable, the flatter landscape for Adam $v_{0,rnd}$ explains its superior testing accuracy. Moreover, as discussed in Appendix C.5, there is no linear mode connectivity between the solutions found by Vanilla Adam and those found using the proposed initialization strategy.

**Ablation study.** The scaling factor $\sigma$ is a key hyperparameter in the proposed initialization method Equations (10) and (11). To evaluate the impact of $\sigma$, we conducted an ablation study on the CIFAR-10 image classification task, as summarized in Table 6. The results show that

Table 6: Impact of $\sigma$ on CIFAR-10 Test Accuracy.

| $\sigma$ | 0 | 0.1 | 1 | 10 | 100 | 1000 |
|---|---|---|---|---|---|---|
| $v_{0,rnd}$ | 95.25 | 95.45 | 95.74 | **95.89** | 95.87 | 95.84 |
| $v_{0,data}$ | 95.25 | 95.70 | **96.02** | 95.92 | 95.85 | 95.72 |

for a wide range of $\sigma$ values, such as $\sigma \in [1, 1000]$, the performance consistently outperforms zero initialization. This highlights the robustness and tuning-friendly nature of the proposed approach, as it achieves stable improvements across different $\sigma$ settings.

**Comparison between warmup.** The warmup technique [17, 47] is a widely used approach to mitigate the sign-descent behavior observed in Adam's early steps. However, it introduces additional hyperparameters, such as scheduling parameters, which require careful tuning. Moreover, warmup often involves several initial training steps during which network parameters are not effectively updated. In contrast, our method directly addresses the aggressive sign-descent issue by initializing $v_0$ with non-zero values, thereby eliminating the need for a warmup phase. As shown in Appendix C.4, the comparison experiments demonstrate that random initialization of $v_0$ stabilizes the training process effectively, without requiring extra hyperparameter tuning or wasted iterations.

## 4. Conclusion

In this work, we revisited the initial steps of adaptive gradient optimization methods, focusing on the instability caused by the sign-descent behavior during early iterations. To address this issue, we proposed two simple yet effective approaches: data-driven initialization and random initialization of the second-moment estimate $v_0$. Our empirical results demonstrate that these initialization strategies significantly enhance the performance and stability of several adaptive gradient optimization methods, including Adam, particularly in challenging tasks such as training Transformer models.

**Acknowledgments**

This research is supported by the National Science Foundation (NSF) under Grant No. 2144489.

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

# A. Additional Details about Second-order Moment Initialization

## A.1. Linear Loss

To simplify the analysis, we consider the RMSprop update rule (ignoring $\epsilon$) for a linear loss. The update for the parameter $\theta_t$ can be expressed as:

$$\mathbf{E}[\Delta\theta_t] = -\alpha\mathbf{E}\left[\frac{g_t}{\sqrt{v_t}}\right] \tag{15}$$

Using a Taylor expansion of $1/\sqrt{v_t}$ around $\mathbf{E}[v_t]$, we approximate:

$$\frac{1}{\sqrt{v_t}} \approx= \frac{1}{\mathbf{E}[v_t]} - \frac{1}{2\mathbf{E}[v_t]^{\frac{3}{2}}}(v_t - \mathbf{E}[v_t]) \tag{16}$$

Substituting this into the expectation, we have:

$$\mathbf{E}[\Delta\theta_t] \approx -\alpha\left(\frac{\mathbf{E}[g_t]}{\sqrt{\mathbf{E}[v_t]}} - \frac{\mathbf{E}[g_t(v_t - \mathbf{E}[v_t])]}{2\mathbf{E}[v_t]^{\frac{3}{2}}}\right) \tag{17}$$

Considering $\mathbf{E}[g_t] = \bar{g}$, and that $g_t$ and $v_t - \mathbf{E}[v_t]$ are uncorrelated, we have: $\mathbf{E}[g_t(v_t - \mathbf{E}[v_t])] = \mathbf{E}[g_t] \cdot \mathbf{E}[v_t - \mathbf{E}[v_t]] = 0$. This simplifies the expression to:

$$\mathbf{E}[\Delta\theta_t] \approx -\alpha\frac{\bar{g}}{\sqrt{\mathbf{E}[v_t]}} \tag{18}$$

$$\approx -\alpha\frac{\bar{g}}{\sqrt{\beta_2^t v_0 + (1 - \beta_2^t)(\bar{g}^2 + \sigma^2 I)}} \tag{19}$$

**Case 1: vanilla Adam ( $v_0 = 0$).** When $v_0 = 0$, the update becomes:

$$\mathbf{E}[\Delta\theta_t] \approx -\alpha\frac{\bar{g}}{\sqrt{(1 - \beta_2^t)(\bar{g}^2 + \sigma^2 I)}} \tag{20}$$

In this setting, the denominator is initially small due to $(1 - \beta_2^t)$ approaching $0$ as $t \to 0$. The small denominator leads to excessively large initial updates, particularly when $\bar{g}$ is small or $\sigma^2$ is large. This instability can cause erratic optimization behavior, especially in the early stages of training.

**Case 2: non-zero initialization ( $v_0 = \bar{g}^2 + \sigma^2 I$).** When $v_0 = \bar{g}^2 + \sigma^2 I$, the update becomes:

$$\mathbf{E}[\Delta\theta_t] \approx -\alpha\frac{\bar{g}}{\sqrt{\bar{g}^2 + \sigma^2}}. \tag{21}$$

In this setting, the denominator is well-scaled from the start, incorporating the correct statistical variance. This prevents excessively large updates during early iterations, ensuring better stability. The step sizes remain consistent across iterations, aligning with the principles of adaptive gradient methods. Additionally, the incorporation of gradient statistics $\bar{g}^2 + \sigma^2 I$ ensures that $v_t$ adapts appropriately to the local geometry of the loss function, such as the Hessian information. For a linear loss, this stabilization leads to smoother convergence, providing a more robust optimization process. It is worth noting that the above analysis can be readily extended to other adaptive gradient methods, such as Adam.

## A.2. Revisiting Previous Works on Stabilizing the Initial Steps of Adam

**Warmup.** The warmup technique [17, 47] implicitly adjusts the initialization of the second-moment estimate $v$ by employing a smaller learning rate during the initial steps. While the optimizer's state updates normally, the parameter changes are minimal due to the extremely small learning rate. This approach effectively mitigates the sign-descent behavior observed in Adam's early steps. However, warmup introduces additional hyperparameters (e.g., the scheduler) that require careful tuning and necessitates several steps of training where the network parameters are not effectively updated.

This can be inefficient, particularly in resource-constrained settings. In contrast, our method directly addresses the aggressive sign-descent issue by initializing $v_0$ with non-zero values, eliminating the need for a warmup phase. Our experimental results demonstrate that random initialization of $v_0$ stabilizes the training process effectively, without requiring extra tuning or wasted iterations.

**RAdam**. RAdam [11] avoids the sign-descent issue by behaving like SGD [47] during the initial steps. This is achieved by introducing a rectification term, dynamically adjusting the optimizer's behavior to stabilize updates in the early iterations. While RAdam successfully addresses initial-step instability, it adds complexity to the optimization process through the computation of the rectification term. In contrast, our approach provides a simpler and more intuitive solution by directly adjusting the initialization of the moment estimates, without modifying the core algorithm or introducing additional dynamic terms.

**AdaBound**. AdaBound [12] tightly bounds the update size during the initial steps, preventing excessively large updates caused by sign-descent behavior. However, this approach introduces dynamic bounds that require careful tuning of the bounding functions, adding additional complexity to the optimization process. Our initialization strategy simplifies this issue by stabilizing updates without the need for dynamic bounds, making it a more efficient and practical alternative.

**AdaBelief**. AdaBelief [13] reduces the impact of initial sign-descent behavior by refining the variance estimation, leading to more reliable adaptive learning rates. However, this comes at the cost of increased computational complexity due to the need for precise variance estimation. By contrast, our method provides stability during the initial steps without additional computational overhead, offering a straightforward alternative to improve early optimization dynamics.

Our initialization strategy can be seamlessly integrated into existing methods, such as RMSprop, AdamW, RAdam, AdaBound, AdaBelief, and even Warmup. By addressing the aggressive sign-descent behavior directly through non-zero initialization of $v_0$, we enhance the stability of these optimizers in their early steps. Importantly, this random initialization incurs no extra computational costs and avoids the need for additional hyperparameter tuning.

# B. Initialization Algorithms for Adaptive Gradient Methods

In this section, we present the pseudocode for the proposed initialization methods for adaptive gradient algorithms, implemented in PyTorch. Algorithm 1 outlines the pseudocode for random initialization ($v_{0,\mathrm{rnd}}$), while Algorithm 2 details the pseudocode for data-driven initialization ($v_{0,\mathrm{data}}$). It is important to note that the second-order moment for network biases is not initialized and remains zero. Only weight matrices with two or more dimensions are initialized with non-zero values.

Algorithm 1 : PyTorch Pseudocode for Random Initialization $v_{0,rnd}$

```
# optim: PyTorch optimizer (e.g., Adam)
# sigma: Scaling factor

for theta in optim.parameters():
    fan_in, fan_out = theta.size(1), theta.size(0)
    if theta.dim() > 2:
        receptive_field = torch.prod(torch.tensor(theta.shape[2:])).item()
        fan_in = fan_in * receptive_field
        fan_out = fan_out * receptive_field
    chi = torch.randn_like(theta)  # Sample from standard normal
    v_0 = (sigma / (fan_in + fan_out)) * (chi ** 2)  # Compute scaled chi^2
    optim.state[theta]['exp_avg_sq'] = v_0  # Assign to optimizer state
```

Algorithm 2 : PyTorch Pseudocode for Data-driven Initialization $v_{0,data}$

```python
# model: PyTorch model
# data_loader: Data loader for the dataset
# criterion: Loss function (e.g., CrossEntropyLoss)
# optim: PyTorch optimizer (e.g., Adam)
# sigma: Scaling factor

# Accumulate gradient statistics
grad_sum = defaultdict(torch.zeros_like)
grad_sq_sum = defaultdict(torch.zeros_like)
for inputs, targets in data_loader:
    outputs = model(inputs)  # Forward pass
    loss = criterion(outputs, targets)  # Compute loss
    model.zero_grad()
    loss.backward()  # Backward pass
    for param in model.parameters():
        if param.grad is not None:
            grad_sum[param] += param.grad
            grad_sq_sum[param] += param.grad ** 2

# Compute expected value (mean) and variance of gradients
num_samples = len(data_loader.dataset)
for param in model.parameters():
    grad_mean = grad_sum[param] / num_samples
    grad_var = (grad_sq_sum[param] / num_samples) - grad_mean ** 2
    # Compute v_0 using the equation: v_0 = sigma * (E[g]^2 + VAR[g])
    v_0 = sigma * (grad_mean ** 2 + grad_var)
    optim.state[param]['exp_avg_sq'] = v_0
```

# C. Additional Details of Experiments

## C.1. Experimental Setting

We empirically evaluate the performance of the proposed data-driven initialization (Equation (10)) and random initialization (Equation (11)) strategies across several widely-used adaptive gradient optimization methods. These include SGD with momentum (SGDM) [35, 36], Adam [4], AdamW [37], AdaBound [12], RAdam [11], and AdaBelief [13]. Each optimizer is tested using its standard initialization ($v_0 = 0$) as the baseline, which is then compared against the proposed strategies $v_{0,data}$ and $v_{0,rnd}$. Following experimental protocols established in prior works [11, 13, 14], we perform thorough hyperparameter tuning for learning rate, $\beta_1$, $\beta_2$, and $\epsilon$. To ensure statistical robustness, each experiment is repeated with five random seeds, and we report the mean results along with standard deviations. For data-driven initialization, gradient statistics are computed using 5,000 random samples prior to training, with the scaling factor set to $\sigma = 1$. For random initialization, the scaling factor is set to $\sigma = 100$, demonstrating the tuning-friendly nature of the proposed approach.

**Image Classification with CNN.** We evaluate the ResNet-34 [32] architecture on the CIFAR-10 image classification dataset [38]. Each model is trained for 200 epochs with a batch size of 128, and the learning rate is decayed by a factor of 0.2 at epochs 60, 120, and 160. Label smoothing [48] with a smoothing factor of 0.1 is applied. In addition to CIFAR-10, we perform experiments on the ImageNet ILSVRC 2012 dataset [39] using ResNet-18 as the backbone network. Each optimizer is executed for 100 epochs with a cosine annealing learning rate schedule, which has demonstrated superior performance compared to step-based decay strategies [49]. For SGD, we use the momentum factor of 0.9, a common default setting [32], with a tuned learning rate of 0.1. For adaptive gradient methods (Adam, AdamW, RAdam, AdaBound, AdaBelief), we use the learning rate of 0.001, $\beta_1 = 0.9$, $\beta_2 = 0.999$, and $\epsilon = 10^{-8}$.

**Language Modeling with LSTM.** We evaluate a 2-layer LSTM [34] on the Penn Treebank dataset [40]. Models are trained for 200 epochs with a batch size of 20, and the learning rate is reduced by

a factor of 0.1 at epochs 100 and 145. For SGD, we use a learning rate of 30 and a momentum factor of 0.9. Adam, AdamW, AdaBound, and AdaBelief use a learning rate of 0.01, while RAdam uses a learning rate of 0.001. All adaptive methods are configured with $\beta_1 = 0.9$ and $\beta_2 = 0.999$.

**Neural Machine Translation with Transformer.** We experiment with a small Transformer model [17] implemented using the Fairseq package [41] on the IWSLT'14 German-to-English machine translation dataset. The model is trained with a length penalty of 1.0, a beam size of 5, and an initial warmup step size of $10^{-7}$. Training is conducted for 55 epochs, and results are reported as the average of the last 5 checkpoints. Adaptive learning methods use a learning rate of 0.0015. Adam, AdamW, AdaBound, and AdaBelief are configured with $\beta_1 = 0.9$, $\beta_2 = 0.98$, while RAdam uses $\beta_1 = 0.9$, $\beta_2 = 0.999$.

**Image Generation with GAN.** We evaluate a deep convolutional GAN (DCGAN) [43] on the CIFAR-10 image generation task. Both the generator and discriminator networks use CNN architectures. Models are trained for 200,000 iterations with a batch size of 64. Learning rate is fixed at 0.0002 for both the generator and discriminator across all optimizers. All other hyperparameters are set to their default values for fair comparison.

## C.2. Additional Results for the Saddle Objective Function

We provide the additionnal results for the saddle objective function discussed in Section 3.1. The final converged parameter values for each method are summarized in Table 7. These results highlight that the proposed method achieves the lowest loss among all optimization techniques, underscoring its effectiveness in handling this optimization task.

Table 7: Final converged parameter values for different optimization methods.

| Adam (vanilla) | Adam + warmup | Adam ($v_{0,\text{rnd}}$) | SGD |
|---|---|---|---|
| -0.96 | 0.01 | $1 \times 10^{-7}$ | $9 \times 10^{-7}$ |

## C.3. Language Modeling with 3-Layer LSTM

We evaluate a 3-layer LSTM network on the Penn Treebank dataset [40]. The test perplexity results are summarized in Table 8. Similar to the findings with the 2-layer LSTM, the proposed initialization strategies provide additional performance gains for adaptive gradient optimization methods.

Table 8: Test perplexity ↓ of 3 Layer LSTM on Penn Treebank dataset dataset.

| Optimization | SGD | Adam | AdamW | AdaBound | RAdam | AdaBelief |
|---|---|---|---|---|---|---|
| Vanilla $v_{0,0}$ | 63.52±0.16 | 64.10±0.25 | 69.91±0.20 | 63.52±0.11 | 70.10±0.16 | 61.33±0.19 |
| $v_{0,rnd}$ | - | 62.68±0.19 | 66.43±0.18 | 62.75±0.11 | 68.05±0.16 | 61.29±0.15 |
| $v_{0,data}$ | - | 62.46±0.20 | 66.38±0.18 | 62.07±0.11 | 68.14±0.16 | **60.70±0.14** |

## C.4. Comparison between Warmup and Proposed Initialization

The warmup strategy, which begins with a small learning rate and incrementally increases it to the standard value, is widely used in neural network training to stabilize the training process. This approach serves a similar purpose to the proposed initialization strategy. However, warmup often requires several initial training steps during which network parameters are not effectively updated. In contrast, our method directly addresses the aggressive sign-descent issue by initializing $v_0$ with non-zero values, eliminating the need for a warmup phase. To illustrate the superiority of the proposed initialization strategy, we conducted experiments comparing it to the warmup approach.

The test accuracy of ResNet-34 on the CIFAR-10 image classification dataset is presented in Table 9. While the warmup strategy slightly improves accuracy compared to vanilla Adam ($v_{0,0}$), the proposed initialization methods, $v_{0,\text{rnd}}$ and $v_{0,\text{data}}$, outperform Adam with warmup. This improvement

Table 9: Test accuracy ↑ for ResNet-34 on CIFAR-10: warmup vs. proposed method.

| Adam (vanilla) | Adam + warmup | Adam ($v_{0,\text{rnd}}$) | Adam ($v_{0,\text{data}}$) |
|---|---|---|---|
| 95.25±0.11 | 95.31±0.09 | **95.87±0.09** | **96.02±0.09** |

occurs because the warmup strategy starts with a very small learning rate, which inefficiently utilizes gradients to update parameters, primarily updating $v_t$ instead. In contrast, our method, with a better initialization of $v_0$ and a larger learning rate, effectively updates parameters from the beginning, as shown in Figure 6. Moreover, our approach achieves an even better final convergence performance.

The test perplexity of a 2-layer LSTM on the Penn Treebank language modeling task is shown in Table 10. The proposed initialization methods, $v_{0,\text{rnd}}$ and $v_{0,\text{data}}$, achieve lower (better) perplexity compared to Adam with warmup. The FID score for the Image Generation with GAN task is presented in Table 11. Similarly, the proposed initialization methods demonstrate superior image generation quality compared to Adam with warmup, as reflected by their lower FID scores. For the Neural Machine Translation task using Transformers on the IWSLT'14 DE-EN dataset, the results are summarized in Table 12. Notably, the default setup for this task employs a warmup strategy; without it, Transformers trained with Adam fail to converge. This behavior, which aligns with observations in Figure 1(a) and previous studies [11],

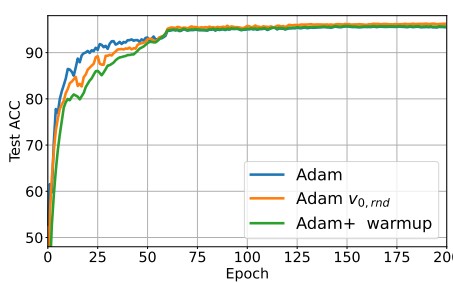

Figure 6: Comparison of Vanilla Adam, Adam with warmup, and Adam $v_{0,rnd}$ on CIFAR-10 image classification task.

highlights the critical role of initialization. However, with the proposed non-zero initialization strategies, $v_{0,\text{rnd}}$ and $v_{0,\text{data}}$, the Transformer successfully converges, as evidenced by the training curves in Figure 1(a). Furthermore, when combined with warmup, these proposed initialization methods outperform the default Adam with warmup strategy, achieving better overall performance.

Table 10: Test perplexity ↓ of 2 Layer LSTM on PTB dataset: warmup vs. proposed method.

| Adam (vanilla) | Adam + warmup | Adam ($v_{0,\text{rnd}}$) | Adam ($v_{0,\text{data}}$) |
|---|---|---|---|
| 67.11±0.20 | 67.12±0.19 | **66.70±0.17** | **66.37±0.17** |

Table 11: FID score ↓ of GAN on CIFAR-10 dataset dataset: warmup vs. proposed method.

| Adam (vanilla) | Adam + warmup | Adam ($v_{0,\text{rnd}}$) | Adam ($v_{0,\text{data}}$) |
|---|---|---|---|
| 54.22±4.21 | 55.87±4.02 | **48.60±3.19** | **47.02±3.20** |

## C.5. Linear Mode Connectivity Analysis for Adam Initialization

The superiority of Adam with non-zero initialization over vanilla Adam can be linked to the ruggedness of the loss landscape. Vanilla Adam, with zero initialization of the second moment estimate, often takes overly aggressive steps during the early stages of optimization. This behavior increases the likelihood of convergence to suboptimal local minima or saddle points, especially in complex, non-convex loss landscapes commonly encountered in deep neural network training. The ruggedness of such landscapes often leads to disconnected basins of attraction, where different optimization trajectories result in vastly different local minima.

To illustrate this phenomenon, we employ Linear Mode Connectivity (LMC) [50], which demonstrates that the optima found using the proposed method and vanilla Adam are not linearly connected. This observation implies that initializing the second-order moment differently alters the resulting loss landscape compared to vanilla Adam. We also leverage the concept of Linear Interpolation Instability [50]. Let $\theta_t^{v_0,0}$ and $\theta_t^{v_0,\text{rnd}}$ represent the network parameters at time step $t$,

Table 12: BLEU score ↑ of Transformer on IWSTL'14 DE-EN dataset: warmup vs. proposed method.

| Methods | Adam (vanilla) | Adam ($v_{0,\text{rnd}}$) | Adam ($v_{0,\text{data}}$) |
|---|---|---|---|
| w/o warmup | <10.0 | 31.53±0.24 | 30.74±0.28 |
| with warmup | 30.14±0.39 | **33.71±0.19** | **33.64±0.20** |

optimized using vanilla Adam ($v_{0,0}$) and Adam with the proposed random initialization ($v_{0,\text{rnd}}$), respectively. Both networks share the same architecture, initialization, dataset, and random seeds. Let $\mathcal{E}(\theta)$ denote the test error of a network with weights $\theta$, and define $\mathcal{E}_\alpha(\theta_1, \theta_2) = \mathcal{E}(\alpha\theta_1 + (1-\alpha)\theta_2)$ for $\alpha \in [0, 1]$, representing the test error of a network created by linearly interpolating between $\theta_1$ and $\theta_2$. Furthermore, let $\mathcal{E}_{\text{sup}}(\theta_1, \theta_2) = \sup_\alpha \mathcal{E}_\alpha(\theta_1, \theta_2)$ denote the maximum error along this linear interpolation path, and $\bar{\mathcal{E}}(\theta_1, \theta_2) = \text{mean}(\mathcal{E}(\theta_1), \mathcal{E}(\theta_2))$ represent the average error between $\theta_1$ and $\theta_2$. The error barrier height, which serves as our measure of instability along the linear path, is defined as:

$$\text{instability} = \mathcal{E}_{\text{sup}}(\theta_t^{v_{0,0}}, \theta_t^{v_{0,\text{rnd}}}) - \bar{\mathcal{E}}(\theta_t^{v_{0,0}}, \theta_t^{v_{0,\text{rnd}}}). \tag{22}$$

Figure 7(a) illustrates the test error when linearly interpolating between the converged minima found by the vanilla Adam optimizer ($\theta_t^{v_{0,0}}$) and the proposed randomly initialized Adam optimizer ($\theta_t^{v_{0,\text{rnd}}}$). The results clearly show that the two networks are not linearly connected, as indicated by the instability during interpolation. Figure 7(b) depicts the linear interpolation instability, as defined in Equation (22), measured across different epochs. At the start of training, the networks are identical, but the non-zero initialization of the second-order moment in Adam causes the optimization process to converge to different optima in distinct regions of the loss landscape. Consequently, the solutions found by vanilla Adam and Adam with $v_{0,\text{rnd}}$ are are non linearly connected. This observation is further supported by the distinct loss landscapes shown in Figure 5.

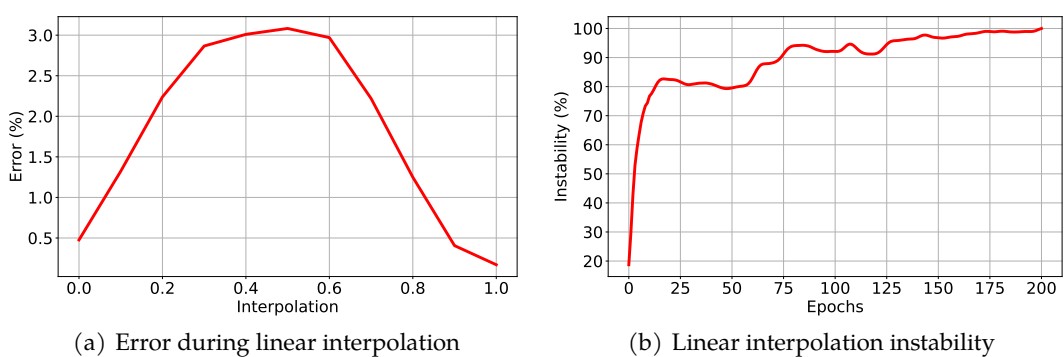

(a) Error during linear interpolation    (b) Linear interpolation instability

Figure 7: Linear Mode Connectivity Analysis. (a) Error observed when linearly interpolating between networks trained with different optimizer initialization strategies: Vanilla Adam ($\theta_t^{v_{0,0}}$) and the proposed method ($\theta_t^{v_{0,rnd}}$), corresponding to interpolation points 0.0 and 1.0, respectively. (b) Linear interpolation instability, measured over training iterations.

