# OpenReview forum: "Revisiting the Initial Steps in Adaptive Gradient Descent Optimization"
_CPAL.cc/2025/Proceedings_Track — CPAL 2025 (Proceedings Track) Poster_

### Official Review · Reviewer_Kpjm · 2025-01-06

**Rating:** 4
**Confidence:** 4

**Review:**

**Summary**: This work studied the instability of Adam and attributed this instability to the zero initialization of the second-order moment estimation $v_0 = 0$. The authors proposed to fix this issue by initializing the second-order moment estimation with non-zero values using either data-driven or random initialization.

**Clarity**: the presentation of this work is very clear.

**Evaluation and Question**: I think it is good to identify the cause of the instability of Adam as the very large first step, although I feel this result is less exciting. Isn't this can be easily fixed by using very small learning rate at the very beginning and then gradually tuning up the learning rate by using some learning rate schedule? Then what is the point of modifying the initialization of the second-order moment estimation?

---

### Official Review · Reviewer_7tf3 · 2025-01-10
**Review of submission 70**

**Rating:** 8
**Confidence:** 4

**Review:**

**Strengths**

1. This paper was well written and motivated. The results were clearly explained and the figures easy to interpret.

2. The authors demonstrate the effectiveness of their approach across a range of architectures and tasks. Additionally, the authors show that their approach improves the performance of multiple adaptive optimizers (making the work generally applicable).

3. The intuition provided by the linear loss and the non-convex toy problem were very helpful and made the point especially clear.

4. The overall results (as well as the goals of the work) improve the efficiency of Adam (and other adaptive optimizers) by shedding parsimonious light on the process. This is very much in-line with the goals of CPAL.

5. Figure 2 was a great analysis and very clearly demonstrated the authors points.

6. The ability to explain the success of Adam with a warmup was a nice corollary of this work.

**Weaknesses**

I did not identify any major weaknesses in this work. I do, however, have two suggestions that I think could further improve the impact and clarity of this work.

1. The intuition for why Adam with non-zero $v_0$ outperformed vanilla Adam was clear in the case of the toy problem. Namely, that vanilla Adam converged to a suboptimal local minima/saddle point. I think it could be impactful to connect the results the authors found on the various DNN architectures to the ruggedness of the loss landscapes and how taking a big step initially could lead to a disconnected local minima. Maybe demonstrating that there is not linear mode connectivity (https://arxiv.org/abs/1912.05671) between the solution found with vanilla Adam and the solution found with the author's method could make the connection to the loss landscape even more salient.

2. The Introduction was generally well written and a good motivator for the work the authors pursue. However, the transition from saying that adaptive optimizers have been very successful at training Transformers (lines 23-27) to saying that for Transformers adaptive optimizers can be unstable (lines 28-31) was a little abrupt and felt somewhat contradictory. Providing a little more detail on how both can be true might make the introduction even more motivated.

**Minor points**

1. A few minor typos:
    i. "We show that the zero initialization of the second-order moment as a significant..." (line 54)
    ii. "When $\beta_$ closer to 1..." (line 153)

2. In Fig. 4 the author's proposed method looks like it is being referred to as $v_{0, md}$ not $v_{0, rnd}$.

3. (Very minor) Both Chi-squared and chi-squared are used.

4. (Very minor) I personally like writing fan$_{in}$ and fan$_{out}$ better from an aesthetics perspective.

---

### Official Review · Reviewer_XMt8 · 2025-01-11
**This paper identifies the zero initialization of the second-order moment as a key factor contributing to the instability of vanilla Adam. By introducing novel initialization methods, the authors improve Adam's stability and enhance the performance of several baseline methods across various tasks.**

**Rating:** 7
**Confidence:** 2

**Review:**

## Quality

The quality of the paper is high. It clearly demonstrates a factor causing Adam's instability and provides a reasonable solution through theoretical analysis. The proposed method yields notable improvements.

## Clarity

The paper is well-written and easy to follow. Despite not being an expert in the optimization domain, I had no difficulty understanding its content.

## Originality

The originality of this paper is good. While the idea is simple, it is both practical and impactful.

## Significance

The significance of this paper is moderate. See the third point my cons on the question.

## Pros

1. The paper is clearly written and presents a cohesive narrative.
2. The proposed method is straightforward and effective.
3. The method aligns well with the theoretical analysis provided.

## Cons

1. Equation 8 seems to have a typo.
2. For the random initialization of $v_0$:
    - do you have different initialization for different weight?
    - If the weight matrix could not be formulated as $\mathcal R^{fan in \times fan out}$, how to determine the initialization .For example, a matrix with three axes.
    - Why is (fan in + fan out) instead of other formulas like (fan in * fan out).
3. In the experiment section, the author only compares their method with Adam + Warmup in section 3.1. From the current version of the paper, whether the proposed method outperforms Adam + Warmup is unclear on those benchmarks. I suggest the author add a comparison of  Adam + Warmup in sections 3.2 - 3.5. Outperforming Adam + Warmup will make your method more significant.

---

### Meta-Review · Area_Chair_1HJG · 2025-02-04

**Recommendation:** Accept (Poster)
**Confidence:** 4

**Metareview:**

The paper provides a clear and well-motivated analysis of Adam's instability, attributing it to the zero initialization of the second-order moment estimation and proposing a straightforward yet effective solution. The theoretical analysis aligns well with the proposed method, and the empirical results demonstrate its effectiveness across various architectures and tasks. Overall, I think this is an interesting work and its insights look convincing. I recommend acceptance.

---

### Decision · Program_Chairs · 2025-02-11

Accept (Poster)